# Structure and Anti-Inflammatory Activity Relationship of Ergostanes and Lanostanes in *Antrodia cinnamomea*

**DOI:** 10.3390/foods11131831

**Published:** 2022-06-22

**Authors:** Xin Yang, Xiang Wang, Jiachen Lin, Sophie Lim, Yujia Cao, Siyu Chen, Pingkang Xu, Chunyuhang Xu, Hongling Zheng, Kuo-Chang Fu, Chien-Liang Kuo, Dejian Huang

**Affiliations:** 1Department of Food Science and Technology, National University of Singapore, 2 Science Drive 2, Singapore 117542, Singapore; yangxin@nus.edu.sg (X.Y.); wang_x@u.nus.edu (X.W.); jiachen.lin01@rafflesgirlssch.edu.sg (J.L.); sophielim.r1@gmail.com (S.L.); yujia.cao@u.nus.edu (Y.C.); chen.siyu.chen@u.nus.edu (S.C.); xupingkang@u.nus.edu (P.X.); e0669581@u.nus.edu (C.X.); e0035606@u.nus.edu (H.Z.); 2AgriGADA Biotech Pte Ltd., 8 Eu Tong Sen Street #17–82, The Central, Singapore 059818, Singapore; michael@vital-wellspring.com (K.-C.F.); cl.kcll@gmail.com (C.-L.K.); 3Ph.D. Programme for Aging, College of Medicine, China Medical University, Taichun 40678, Taiwan; 4National University of Singapore (Suzhou) Research Institute, 377 Linquan Street, Suzhou 215123, China

**Keywords:** *Antrodia cinnamomea*, ergostanes, lanostanes, triterpenoids, antcin A, anti-inflammation

## Abstract

*Antrodia cinnamomea* is a precious edible mushroom originating from Taiwan that has been popularly used for adjuvant hepatoprotection and anti-inflammation; however, the chemical principle for its anti-inflammatory activity has not been elucidated, which prevents the quality control of related products. Using the RAW264.7 model for the anti-inflammatory activity assay as a guide, we reported the isolation and structural elucidation of three potent anti-inflammatory compounds from isolated ergostanes (16) and lanostanes (6). Their structures were elucidated on the basis of spectroscopic data analysis including NMR and HR-QTOF-MS. Particularly, the absolute configurations of (25R)-antcin K, (25R)-antcin A, versisponic acid D, and (25R)-antcin C were determined by single crystal X-ray diffraction (XRD). The representative and most promising compound antcin A was shown to suppress pro-inflammatory biomolecule release via the down-regulation of iNOS and COX-2 expression through the NF-κB pathway while the mRNA levels of IL-1β, TNF-α and IL-6 were also decreased. The high dependency on structural variation and activity suggests that there might be special biological targets for antcin A. Our work makes it possible to develop evidence-based dietary supplements from *Antrodia cinnamomea* based on anti-inflammatory constituents.

## 1. Introduction

*Antrodia cinnamomea* is a folk mushroom growing on the inner cavity of *Cinnamomum kanehirae Hay*, a member of the Lauraceae family only found in Taiwan [1,2,3,4]. Naturally, it is grown in the mountain ranges of Miaoli, Kaohsiung, Taitung and Taoyuan [5], and it is commonly known by its Chinese name as “Niu-chang chih” [1]. In its natural habitat, its fruiting body grows very slowly on its host tree, and it is difficult to cultivate in the greenhouse. Thus, the fungus is considered rare and expensive, and is therefore protected by the local government [6]. Historically, *Antrodia cinnamomea* was used by the aborigines as a treatment for food and alcohol intoxication. In recent times, it is sold in capsules and used in traditional Chinese medicine as a treatment for hypertension, diarrhoea, and other inflammatory disorders [7,8,9].

Current studies have shown that the fruiting bodies of *Antrodia cinnamomea* contain anti-oxidative, hepatoprotective, anti-cancer, and anti-inflammatory activities [10,11,12,13,14,15,16,17]. A study by Hsiao et al. [18] revealed that extracts of *Antrodia cinnamomea* were able to protect against oxidative stress by inhibiting nonenzymatic iron-induced lipid peroxidation in rat brain homogenates (IC_50_ = 3.1 mg/mL) and scavenged the stable free radical 1,1-diphenyl-2-picrylhydrazyl (DPPH). Another study by Rao et al. [11] showed the anti-cancer and anti-inflammatory properties of the fruiting bodies of *Antrodia cinnamomea*. The chloroform and methanol extracts were found to inhibit the excess production of inflammatory mediators such as nitric oxide. *Antrodia cinnamomea* extract was able to inhibit the LPS induction of cytokine, iNOS, and COX-2 expression by blocking NF-κB activation, thus reporting the first confirmation of its anti-inflammatory potential in vitro [12]. However, the existing studies report only limited activity in crude extract, which does not allow the establishment of bioactive molecules. Therefore, the objective of this study is to advance our understanding of the phytochemicals found in *Antrodia cinnamomea* that are responsible for its anti-inflammatory activity. The compounds we isolated are shown in Figure 1.

## 2. Results and Discussion

### 2.1. HPLC/LCMS Analysis of Bioactive Compounds from Antrodia cinnamomea

*A. cinnamomea* fruiting bodies were first extracted after using ethanol (95%) for 1 h to obtain crude extract (yield 19%). The crude extract was further fractionated by four solvents with increasing polarity. The hexane fraction yielded 7%, while the diethyl ether fraction gave 46%, showing high anti-inflammatory activity as measured by the RAW264.7 cell model. On the other hand, the ethyl acetate fraction (16%) and butanol fraction (21%) did not show measurable anti-inflammatory activity. Analytical HPLC-MS characterization of each fraction revealed that there were many compounds detected in the four fractions (Figure 2 and Table 1).

To further establish the structural and activity relationship, the hexane fraction and diethyl ether fraction were subjected to semipreparative HPLC isolation for each compound. Benzenoids and their derivatives were identified in the hexane-soluble fraction, while abundant triterpenoids including ergostane and lanostane types were detected in fractions with higher polarity. Ethyl acetate fraction and the Bu fraction shared a very similar HPLC fingerprint where compound **4** (25S-antcin K) and **5** (25R-antcin K) and a pair of C-25 epimers were dominant. The UHPLC/qTOF-MS of these crude fractions was also conducted for structural characterizations. The compounds with isomeric structures were differentiated by high-resolution mass spectrometry (HRMS) and their fragmentation patterns are summarized in Table 1.

### 2.2. Structural Identification of Compounds **1**-**29** Isolated from Antrodia cinnamomea

Compounds **1, 2**, **6**, **7**, **9**, **12** and **13** were tentatively identified as benzenoids based on their MS fragmentations. Compounds **3**, **4**, **5**, **8**, **10** and **15**-**24** were identified as ergostane-type triterpenoids, while compounds **11**, **14**, **25**-**28** were identified as lanostane-type triterpenoids. Ergostanes and lanostane-type triterpenoids could be preliminary distinguished by their characteristic peaks of ^1^H NMR and ^13^C NMR signals, as shown in Figure 3A. The difference in the ^1^H chemical shift in the protons of C-28 in lanostane (0.083) was larger than that in ergostanes (0.03), while their ^13^C NMR signals had little difference (NMR spectra were listed in Appendix A). The protons in C1 (Figure 3B) were located in the chemical shift at around 2.3 to 2.8 in DMSO-*d*_6_ with a doublet of doublets, and a similar coupling pattern was also observed in protons from C2 (Figure 3C). Hydroxyl substituents commonly appeared in C-3 (Figure 3D), C-5, C7 and C14 in different triterpenoids; they could be identified from the chemical shifts ranging from 3.2 to 4.8 as triplets. A doublet (*J* = 13.5 Hz,) was observed for protons on C12 (Figure 3E), which could be easily identified if C12 was not substituted. The signal for protons in C15 (Figure 3G) was split into a triplet by H in the methyl group of C30, and again into doublets by the H of C16, resulting in a ‘triplet of doublets’. C25 (Figure 3F), located between the vinyl group and the carboxyl group had a low file, and the methyl group accounted for the quadlets in a splitting pattern.

25S-Anctin K (**4**) and 25R-Anctin K (**5**) were obtained as a white, amorphous powder. They had similar polarity and could be separated by repeated semi-prep HPLC with different mobile phases. Their molecular formula was determined by HRMS with signals of *m*/*z* at 487.3058 [M-H]^-^ and 487.3063, respectively, which is in accord with the calculated molecular weight value C_29_H_44_O_6_ (487.3065). The UV absorption maximum at 252 nm revealed the presence of an 8(9)-en-11-one moiety, which was confirmed by the carbon resonances at 201.2 (C-11), 154.0 (C-8), and 143.1 (C-9) ppm. The ^1^H NMR and ^13^C NMR spectra of **4** showed the presence of five methyl groups at δH 1.30 (d, *J* = 5.9 Hz, 3H), 1.15 (dd, *J* = 7.1, 3.3 Hz, 3H), 1.05 (s, 3H), 0.87 (d, *J* = 5.6 Hz, 3H), and 0.64 (s, 3H), while the ^13^C NMR spectra displayed 29 signals suggesting that **4** contained a 4-methylergosta-8(9)-ene-7,11-dione skeleton. However, the absolute configuration of **4** and **5** could not be confirmed, although they could be separated by HPLC at different retention times. After trying several solvent systems, the single crystal of **5** was obtained in MeOH:H_2_O = 90:10, and then the absolute configuration of **5** was determined as 25S by its X-ray crystallography data (Figure 4).

(25R)-antcin A (**10**) was obtained as a white powder. HRESIMS analysis established its molecular formula as C_29_H_42_O_4_ with an [M-H]^−^ anion at *m*/*z* 453.3007. In agreement, its ^13^C NMR spectrum displayed 29 carbon signals with *δ*_C_ at 212.3 (C-3), 199.4 (C-11), 175.7 (C-26), 157.8 (C-8), and 138.1 (C-9). The methyl signals for CH_3_-19 (1.25, s), CH_3_-27 (1.15, dd), CH_3_-29 (0.93, d), CH_3_-21 (0.87, d), and CH_3_-18 (0.65, s) suggested the structure of **10** was very similar to that of antcin C. The UV spectrum showed an absorption maximum at 253 nm, indicating the presence of an 8(9)-ene-11-one moiety. The only difference between antcin A and C was the hydroxyl group in C-6, which was supported by ^1^H NMR and ^13^C NMR. The absolute configuration of **10** was determined to be 25R by its X-ray crystallography data (CCDC# 2099280). (Figure 3).

Versisponic acid D (**11**) was isolated as a white powder. It had an [M-H]^−^ ion at *m*/*z* 527.3738 (calcd 527.3742) by HRESIMS analysis that corresponded to the molecular formula of C_32_H_50_O_5_, indicating that **11** is a triterpenoid. The chemical shift difference between H-28^a^ (δH 1.13, d, *J* = 6.4 Hz) and H-28^b^ (δH 1.13, d, *J* = 6.4 Hz) in **11** was significantly larger than that in **4**, **5** and **10**, indicating that **11** belongs to lanostane-type triterpenoids rather than the ergostane type. Its UV absorption maximum at 249 nm suggested the presence of an 8(9)-en-11-one moiety, which was confirmed by the carbon resonances at 26.26 (C-7), 132.40 (C-8), 135.80 (C-9), and 20.533 (C-11). The colourless needle single crystals were obtained from MeOH and acetone mixture (1:1), the absolute configuration of **11** was confirmed to be versisponic acid D by its X-ray crystallographic data (CCDC# 2099279) (Figure 3) for the first time.

(25R)-antcin C (**22**) and (25S)-antcin C (**23**) were identified as a pair of ergostane epimers with the same molecular formula of C_29_H_42_O_5_ by HRMS peaks at *m*/*z* 469.3319 and 469.2953, respectively. Their UV and NMR spectra were identical, although they showed different high-performance liquid chromatography (HPLC) retention. The ^1^H and ^13^C NMR spectra of **22** almost overlapped with those of **4**, except for 3-hydroxyl in **4** instead of 3-ketone in **22**, and one more 4-hydroxyl in **4**. The absolute configuration of **22** was determined to be 25R by its X-ray crystallography data (CCDC#2099281) (Figure 3).

### 2.3. Evaluation of Anti-Inflammatory Activity of Major Compounds

The compounds with high abundancy were selected for the anti-inflammatory test using murine macrophage RAW 264.7 cells. Compounds with concentrations of 50 μM and 100 μM were tested preliminarily for their anti-inflammatory activities (Figure 4). Among these, **2** (H6), **8** (antcin G), **10** (25R-antcin A), **11** (versisponic acid D), **14** (eburicoic acid), **24** (antcamphin P/Q), **27** (15α-acetyl-dehydrosulphurenic acid), **28** (dehydroeburicoic acid) and **29** (structure unknown) exhibited significant anti-inflammatory activity compared to LPS-treated cells. Most bioactive compounds belong to the triterpenoid family, especially lanostane-type triterpenoids. All compounds tested did not show any significant cytotoxicity under the concentrations tested for their activity (Figure 5A). Remarkably, **4** (25S-antcin K) exhibited growth-promoting effects in contrast with the control cells.

Notably, **10** and **11** (10 μM) seemed to be the most conspicuous in relieving the inflammation stimulated by LPS (Figure 5B), and the half-inhibitory concentrations (IC_50_) of **10** and **11** were determined to be 19.61 ± 0.8 (Figure 5C) and 17.16 ± 1.0 (Figure 5D), respectively.

Antcins are dominant bioactive compounds found in *A. cinnamomea,* but only antcin A (**10**), a steroid-like compound, showed measurable anti-inflammatory activity. Its structure bears close similarity with that of glucocorticoid. Hence, it is conceivable that antcin A could bind the same biological target as the glucocorticoid receptor and activate anti-inflammatory responses or suppress inflammatory reaction [19]. In contrast, antcin C (**22** and **23**), H (**19** and **20**) and K (**4** and **5**) had hydroxyl at C-7 which prevented them from binding to the glucocorticoid receptor. Additionally, antcin K showed anti-inflammatory activity by inhibiting NF-κB activation and interleukin-8 production in LPS-induced human gastric cancer AGS cells [20], and we did not observe its activity in NO inhibition. The derivative of antcin K, methyl antcinate K (**17**), showed activity in NO suppression in LPS- and IFN-γ-induced BV-2 cell lines, which only comprised 10% and 5% of the NOS inhibition activity of L-NAME [21]. The other type of extracts, lanostanes, also exhibited inflammation inhibitory activities. For example, eburicoic acid (**14**) and dehydroeburicoic acid (**28**) significantly suppressed λ-carrageenan-simulated paw edema in mice by down-regulating mediators including NO, iNOS, COX-2, TNF-α, and IL-1β. Notably, the dehydrogenation slightly improved the activity of dehydroeburicoic acid [14]. The biological targets for **14** and **28** await to be determined.

## 3. Materials and Methods

### 3.1. General Experimental Procedures

UV–vis spectra were obtained from Shimadzu UV-2101 PC (Shimadzu Corporation, Kyoto, Japan) or Waters 2996 Photodiode Array Detector (PDA) (Waters Corporation, Milford, MA, USA). NMR spectral data were recorded on a Bruker Avance III 500 MHz spectrometer. High-resolution electrospray ionization mass spectrometry (HRESIMS) was performed on A 6530 Accurate-Mass LC–QTOF-MS device (Agilent Technologies, Santa Clara, CA, USA). The ACQUITY Arc System (Waters, Milford, MA, USA) was applied for the analytical determination of bioactive compounds from plant samples while the Waters semi-prep HPLC system with a 2998 PDA Detector was used for product separation.

### 3.2. Plant Material

The fruiting bodies of *Antrodia*
*cinnamomea* were provided by AgriGADA Biotech Pte. Ltd. (Singapore).

### 3.3. Extraction and Isolation

Two hundred grams of air-dried fruiting bodies of *Antrodia cinnamomea* were ground into fine powder using a mini blender (DM-6, Youqi, Taiwan) and extracted with 95% ethanol at ambient temperature (3 × 2.0 L). The slurry was filtered and the filtrate was concentrated under reduced pressure to yield a dark-brown residue of 38.9 g as the crude extract, which was suspended in water (1.0 L) and extracted with hexane (HE), diethyl ether (DE), ethyl acetate (EA), and n-butanol (Bu), sequentially. The fractions were dissolved in DMSO (0.1 g/mL) for the screening of anti-inflammatory activity.

The crude fractions with promising activities were further separated by semi-prep HPLC using the Waters Purification System equipped with a 2998 PDA Detector and a 2707 AUTOSAMPLER. A HPLC column (Luna 5 µM C18(2) 100 Å, 250mm × 10 mm; Phenomenex) was employed with two solvent systems: H_2_O with 0.1% formic acid (A) and MeOH (B). 

### 3.4. Characterization of Isolated Products from Antrodia cinnamomea

^1^H NMR spectra were recorded on commercial instruments operating at 400 and 500 MHz (^1^H) and 100 and 125 MHz (^13^C), respectively. Chemical shifts were reported in ppm from the solvent resonance as the internal standard (DMSO-*d*_6_), δ = 2.50). Spectra were reported as follows: chemical shift (δ ppm), multiplicity (s = singlet; d = doublet; t = triplet; q = quartet; hept = heptet; m = multiplet), coupling constants (Hz), and integration and assignment. ^13^C NMR spectra were recorded on commercial instruments (126 MHz). Chemical shifts were reported in ppm with the solvent resonance as the internal standard (DMSO-d, δ = 39.52). HRMS was recorded on a commercial apparatus (ESI source).

### 3.5. Cell Culture

The murine macrophage RAW 264.7 (TIB-71, ATCC) cells were maintained in high glucose Dulbecco’s Modified Eagle Medium (DMEM, Hyclone™, cytiva Pte. Ltd., Singapore), supplemented with 10% foetal bovine serum (FBS, Hyclone™, cytiva Pte. Ltd., Singapore) and 1% penicillin–streptomycin antibiotics (100 IU/mL penicillin and 100 µg/mL streptomycin, Gibco™, ThermoFisher Scientific Pte. Ltd., Singapore), and culture medium was changed every 2 days. Next, 25,000 cells were seeded into 96-well plates for nitric oxide production and the cytotoxicity test, and 750,000 cells were seeded into 6-well plates for protein and mRNA extraction. After cells were incubated at 37 °C with 5% CO_2_ overnight, they were pre-treated with crude fractions or isolated pure compounds for 4 h, and with (for nitric oxide production) or without (for cytotoxicity) 100 ng/mL lipopolysaccharide (LPS, *Escherichia coli* serotype 055:B5, Sigma Aldrich, Singapore) for another 20 h.

### 3.6. Cytotoxic Activities

Cytotoxicity was characterized by cell viability via Cell Counting Kit-8 (CCK-8 Dojindo, Kumamoto, Japan). After 24 h of treatment, 100 µL CCK-8 in DMEM (1:10) was added to cells following incubation at 37 °C with 5% CO_2_, and light was avoided for 45 min. Afterwards, the absorbance value was measured at 450 nm by a microplate reader (Tecan Infinite F200, Tecan Group Ltd., Singapore). The optical density was converted to the percentage of the control group, and all experiments were performed in three independent tests.

### 3.7. Nitric Oxide (NO) Inhibitory Assay

The effects of the compounds isolated from the *A. cinnamomea* extracts on NO production were measured indirectly by the analysis of nitrite levels in cell culture medium using the Greiss reagent assay (Promega, Singapore). Briefly, 50 µL of cell supernatant from *A. cinnamomea* extracts treated with RAW 264.7 cells were collected for NO production measurements following the manufacturer’s protocol. Absorbance value was measured on the microplate reader at 540 nm. The NO concentration was quantified using the nitrite standard curve (1.56–100 µM) and results were expressed as a percentage of inhibition relative to the LPS. Each treatment was carried out in triplicate. IC_50_ represents the concentrations at which 50% of the NO was inhibited by the extracts.

### 3.8. Statistical Analysis

Descriptive statistical analyses were performed using GraphPad Prism 8.0.2 for calculating the means and the standard error of the mean. Results are expressed as the mean ± standard deviation (SD). Data comparison was analysed by one-way analysis of variance (ANOVA). A Dunnett correction was selected for the post hoc test. *p** < 0.05 was considered statistically significant.

Research manuscripts reporting large datasets that are deposited in a publicly available database should specify where the data have been deposited and provide the relevant accession numbers. If the accession numbers have not yet been obtained at the time of submission, please state that they will be provided during review. They must be provided prior to publication.

Interventionary studies involving animals or humans, and other studies that require ethical approval, must list the authority that provided approval and the corresponding ethical approval code.

## 4. Conclusions

Taken together, we have established the molecular basis for the anti-inflammatory activity of *Antrodia cinnamomea,* and identified that the less abundant terpenoids (**4** and **5**) are the active compounds. The anti-inflammatory activity of terpenoids is highly related to their structures. For lanostane-type terpenoids (with double bonds between C8 and C9), the substituents on C15 seemed to be important in the anti-inflammatory activities; compound 25 and compound 26, which had a hydroxyl group on C15 of the backbone, showed significantly lower activities than the rest of the compounds. While ergostane in type, the functional group (hydroxyl group, ester group, and ketone) occurring on C7 of the backbone was found to have a negative impact on anti-inflammatory activity. Our work serves as a foundation for the in-depth study of the health promotion activity of this treasured mushroom.

## Figures and Tables

**Figure 1 foods-11-01831-f001:**
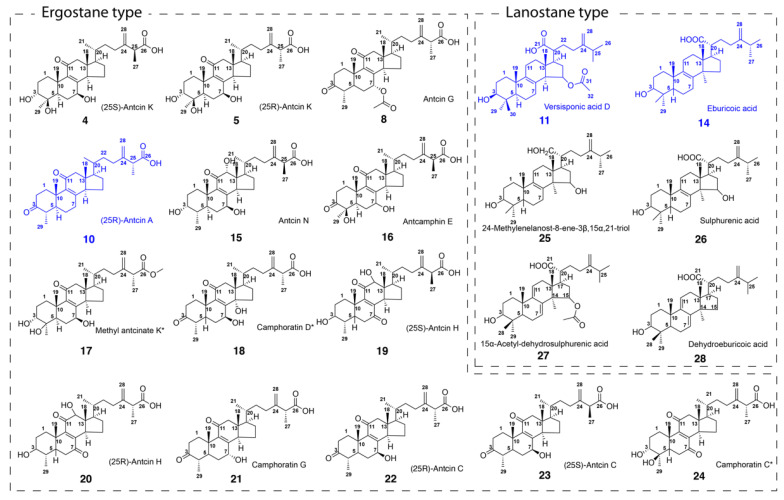
Structures of phytochemicals isolated from cultured *Antrodia cinnamomea* fruiting bodies. The blue ones are potent anti-inflammatory agents. Note: the numbers correspond to the peak numbers in the HPLC chromatogram shown in Figure 2.

**Figure 2 foods-11-01831-f002:**
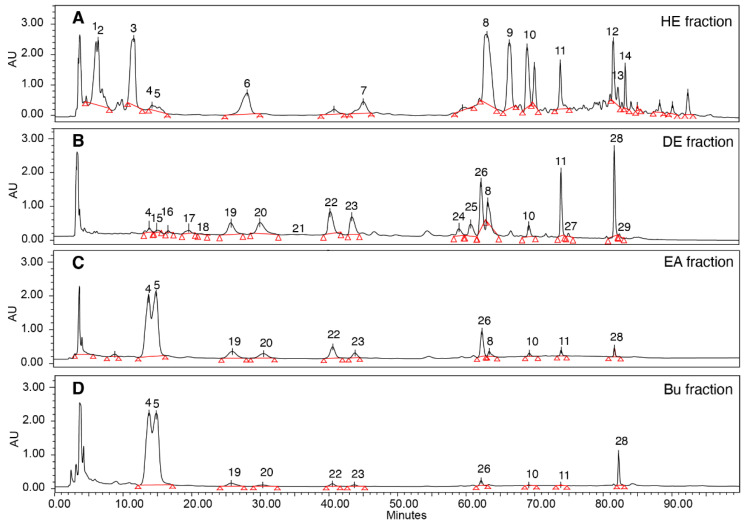
HPLC profiling and bioactivities of *Antrodia cinnamomea* fruiting bodies from different fractions. (**A**) Hexane; (**B**) diethyl ether; (**C**) ethyl acetate; and (**D**) n-butanol fractions. (HE: hexane; DE: diethyl ether; EA: ethyl acetate; Bu: n-BuOH).

**Figure 3 foods-11-01831-f003:**
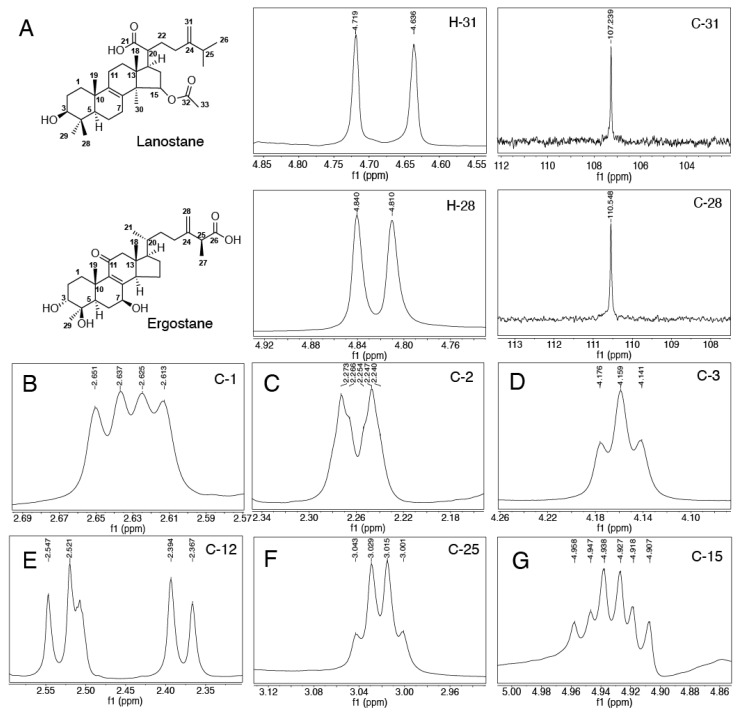
Structural characterization of ergostane-type and lanostane-type triterpenoids. (**A**) ^1^H NMR resonance for H−3 and ^13^C NMR resonance for C-3. The characteristic NMR signal of C1 of (25S)-Antcin K is shown in (**B**), C2, C3, C12, C25 and C15 in (**C**), (**D**), (**E**), (**F**) and (**G**).

**Figure 4 foods-11-01831-f004:**
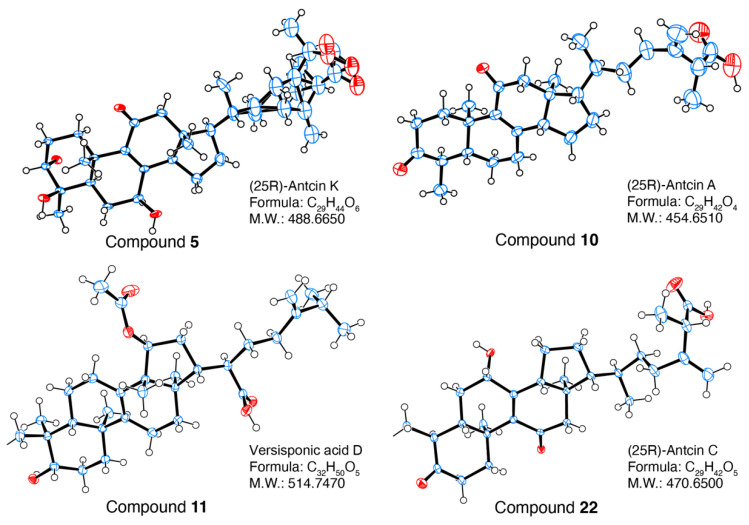
ORTFP drawing of **5**, **10**, **11** and **22**.

**Figure 5 foods-11-01831-f005:**
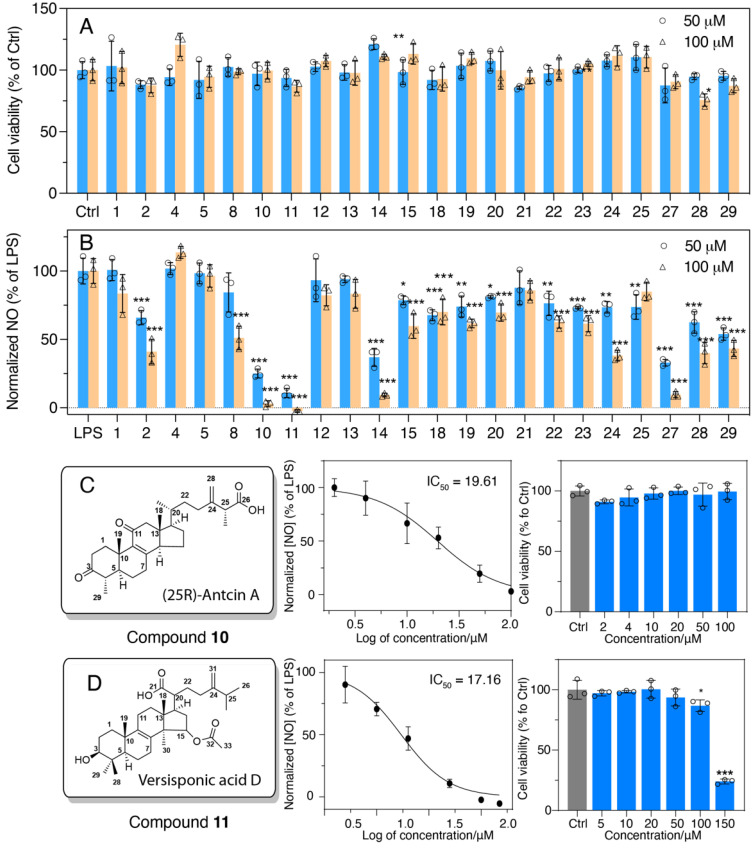
Anti-inflammatory activities of bioactive compounds isolated from *Antrodia cinnamomea*. (**A**) Cell viability of cells treated only with compounds **1**-**29**. Dose–response curve of normalized NO with the treatment of 100 ng/mL LPS with **10**. (**B**) Nitric oxide production of RAW 264.7 cells treated with 100 ng/mL LPS and compounds **1**-**29** isolated from *Antrodia cinnamomea*. (**C**) Compound **10** and (**D**) **11** and their cytotoxicity. Results are shown as mean ± S.D, n = 3, *p** < 0.05, *p*** < 0.01, *p**** < 0.001 vs. LPS or control.

**Table 1 foods-11-01831-t001:** Characterization data of phytochemicals in *A. cinnamomea* by UHPLC/qTOF-MS.

No	t_R_	Max	Formula	[M-H]^−^			MS/MS of [M−H]^−^	Identification
	(min)	(nm)		Pred.	Meas.	Δ(ppm)		
1	6.355	230	C_10_H_14_O_4_	197.0819	197.085	0.0011	-	2,3,4-trimethoxy-6-methylphenol
2	6.421	230	C_24_H_19_Na_2_O_12_^-^	545.0677	544.9673	0.1004	118, 168, 396, 480	Benzenoid (tentative)
3	11.564	276	C_29_H_44_O_6_	487.3065	487.3061	0.0004	119, 247, 407, 425, 443	Unknown
4	14.23	252	C_29_H_44_O_6_	487.3065	487.3058	0.0007	259, 271, 342, 407, 443, 460	(25S)-Antcin K
5	14.988	252	C_29_H_44_O_6_	487.3065	487.3063	0.0002	259, 271, 342, 407, 443, 460	(25R)-Antcin K
6	28.088	284	C_29_H_29_O_9_	485.1218	485.2904	0.1686	272, 301, 425	Benzenoid (tentative)
7	45.031	265	C_25_H_23_O_10_	483.1297	483.2762	−0.1465	193, 287, 337, 411, 421, 439	Benzenoid (tentative)
8	63.086	250	C_31_H_44_O_6_	511.3065	511.3425	0.036	395, 407, 451, 465	Antcin G
9	68.933	251	C_26_H_23_O_11_	452.0822	452.0822	0	126, 143, 262, 316	Benzenoid (tentative)
10	66.352	253	C_29_H_42_O_4_	453.301	453.3007	0.0003	123, 257, 337, 393, 411	(25R)-Antcin A
11	70.011	249	C_32_H_50_O_5_	527.3742	527.3737	0.0005	397. 467	Versisponic acid D
12	73.777	244	C_35_H_29_O_9_	593.1812	593.3732	−0.1920	265	Benzenoid (tentative)
13	81.509	242	C_25_H_29_O_7_	467.1348	467.3529	−0.1095	113, 227, 249, 385, 441	Benzenoid (tentative)
14	83.258	245	C_29_H_41_O_5_	469.2959	469.3684	0.0725	113, 227, 281, 299, 385, 441	Eburicoic acid
15	13.935	198	C_29_H_44_O_6_	487.3065	487.3061	0.0004	209, 407, 452	Antcin N
16	15.058	198	C_29_H_42_O_6_	485.2909	485.2903	0.0006	149, 233, 408, 423, 441, 467	Antcamphin E
17	16.676	198	C_30_H_46_O_6_	501.3222	501.2854	0.0368	325, 358, 395, 413, 439, 457, 483	Methyl antcinate K *
18	19.701	198	C_29_H_42_O_6_	485.2909	485.2906	0.0003	137, 247, 289, 341, 407, 423, 441	Camphoratin D *
19	25.78	256	C_29_H_42_O_6_	485.2909	485.2903	0.0006	149, 207, 233, 399, 423	(25S)-Anctin H
20	30.056	256	C_29_H_42_O_6_	485.2909	485.2902	0.0007	193, 233, 397, 441	(25R)-Anctin H
21	34.119	252	C_29_H_42_O_5_	469.2959	469.2957	0.0002	247, 301, 341, 407, 425	Camphoratin G
22	40.26	255	C_29_H_42_O_5_	469.2959	469.3319	0.036	339, 395, 407, 425	(25R)-Antcin C
23	43.403	255	C_29_H_42_O_5_	469.2959	469.2953	0.0006	219, 233, 272, 391, 425	(25S)-Antcin C
24	58.941	273	C_29_H_42_O_6_	485.2909	485.2906	0.0003	233, 275, 397, 412, 423, 441	Antcamphin P/Q
25	60.664	273	C_31_H_50_O_4_	485.3636	485.2903	0.0733	253, 339, 397, 405, 413, 441	24-Methylenelanost-8-ene-3β,15α,21-triol
26	62.16	242	C_31_H_50_O_4_	485.3636	485.2901	0.0735	160, 369, 397, 423, 441	Sulphurenic acid
27	73.801	242	C_33_H_50_O_5_	525.3585	525.3577	0.0008	233, 245, 397, 467	15α-Acetyl-dehydrosulphurenic acid
28	81.553	240	C_31_H_49_O_3_	467.3531	467.3526	0.0005	156, 170, 339, 371	Dehydroeburicoic acid
29	81.965	202	C_30_H_47_O_4_	469.3323	469.3683	0.036	339	Unknown

## Data Availability

Data are contained within the article.

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
