# Peer review of "Structure and Anti-Inflammatory Activity Relationship of Ergostanes and Lanostanes in Antrodia cinnamomea"

_foods, 2022, doi:10.3390/foods11131831_

Round 1
Reviewer 1 Report
After careful review, this manuscript is well designed.
But, there are also some problems.
1. The yield of further fractionated by four solvents need to be given.
2. Please further explain. In figure 2, why did the HPLC profiling of different fractions had the same peak time, just like compound 4 and 5?
3. Some sentences need further adjustment.
Author Response
Response to Reviewer 1 Comments
After careful review, this manuscript is well designed.
We greatly appreciate your positive and contructive comments and have addressed them with our best efforts. Thank you for your time to review our manuscript.
But, there are also some problems.
Point 1: The yield of further fractionated by four solvents need to be given.
Response 1: The yields of fractions have been added in lin 64 to 66.
Point 2: Please further explain. In figure 2, why did the HPLC profiling of different fractions had the same peak time, just like compound 4 and 5?
Response 2: The extraction efficiency of solvent cannot be 100%. So there still have some compounds remained and extracted by following solvents.
Point 3: Some sentences need further adjustment.
Response 3: Thanks for your comments. We have revised the manuscript very carefully according to your comments. We believe the revised version, can meet the requirement of the journal.

Reviewer 2 Report
In this study the authors analyzed structure-activity relationship of bioactive compounds from the mushroom Antrodia cinnamomea by using modern analytical and in vitro assays. Applying different methods of separation the authors separated several fractions that were analyzed for bioactivitiy by using cell assay and analyzed the structure of present compounds by combination of modern analytical methods (HPLC, LC-MS and NMR). The authors performed fine and detailed analysis of compounds present in the extracts and tested their bioactivity to establish the active principles. It would be helpful if in the cell assay also had positive control besides the control.
The topic is relevant to the field because it reports the active compounds from the ethno medicine and in a way confirms some of claims related to it.
The authors designed study well, its analytical part and bioactivity (in vitro studies – cell assays) part. The obtained data is presented in a good and clear form even though there needs to be some minor corrections.
The conclusions are appropriate and based on the obtained data.
The references are appropriate and sufficient.
There should be some corrections:
Lines 50- 51: it seems a word activity is missing there – limited activity
Line 86: Compounds
Line 167: and (not needed)
Fig.5 - in figure legend A is nitric oxide production and B cell viability. In graphs it is opposite (switched) – please correct
Line 211: how long was the extraction (in minutes) – please add this information
Author Response
Response to Reviewer 2 Comments
In this study the authors analyzed structure-activity relationship of bioactive compounds from the mushroom Antrodia cinnamomea by using modern analytical and in vitro assays. Applying different methods of separation the authors separated several fractions that were analyzed for bioactivitiy by using cell assay and analyzed the structure of present compounds by combination of modern analytical methods (HPLC, LC-MS and NMR). The authors performed fine and detailed analysis of compounds present in the extracts and tested their bioactivity to establish the active principles. It would be helpful if in the cell assay also had positive control besides the control.
The topic is relevant to the field because it reports the active compounds from the ethno medicine and in a way confirms some of claims related to it.
The authors designed study well, its analytical part and bioactivity (in vitro studies – cell assays) part. The obtained data is presented in a good and clear form even though there needs to be some minor corrections.
The conclusions are appropriate and based on the obtained data.
The references are appropriate and sufficient.
We greatly appreciate your positive and constructive comments that have been addressed in full below. Thank you for your time and efforts to help our publication efforts.
There should be some corrections:
Point 1: Lines 50- 51: it seems a word activity is missing there – limited activity
Response 1: Thanks for your correction. The word has been corrected.
Point 2: Line 86: Compounds
Response 2: Thanks for your correction. The word has been corrected.
Point 3: Line 167: and (not needed)
Response 3: Thanks for your correction. The word correction. It has been corrected.
Point 4: Fig.5 - in figure legend A is nitric oxide production and B cell viability. In graphs it is opposite (switched) – please correct
Response 4: The figure caption has been corrected.
Point 5: Line 211: how long was the extraction (in minutes) – please add this information
Response 5: The extraction time has been added in main text line 62.
